# One Positive Label is Sufficient: Single-Positive Multi-Label Learning with Label Enhancement

**Ning Xu[1], Congyu Qiao[1], Jiaqi Lv[2], Xin Geng[1],\*Min-Ling Zhang[1]**

[1]School of Computer Science and Engineering, Southeast University, Nanjing 210096, China
[2] RIKEN Center for Advanced Intelligence Project, Tokyo 103-0027, Japan
{xning, qiaocy}@seu.edu.cn, is.jiaqi.lv@gmail.com,
{xgeng, zhangml}@seu.edu.cn

## Abstract

Multi-label learning (MLL) learns from the examples each associated with multiple labels simultaneously, where the high cost of annotating all relevant labels for each training example is challenging for real-world applications. To cope with the challenge, we investigate single-positive multi-label learning (SPMLL) where each example is annotated with only one relevant label and show that one can successfully learn a theoretically grounded multi-label classifier on SPMLL training examples. In this paper, a novel SPMLL method named SMILE, i.e., Single-positive MultI-label learning with Label Enhancement, is proposed. Specifically, an unbiased risk estimator is derived, which could be guaranteed to approximately converge to the optimal risk minimizer in fully supervised learning and shows that one positive label of each instance is sufficient to train a model. Then, the corresponding empirical risk estimator is established via recovering the latent soft label as a label enhancement process, where the posterior density of the latent soft labels is approximate to the variational Beta density parameterized by an inference model. Experiments on twelve corrupted MLL datasets show the effectiveness of SMILE over several existing SPMLL approaches. Source code is available at https://github.com/palm-ml/smile.

## 1 Introduction

Multi-label learning (MLL) aims to build a predictive model to assign a set of relevant labels for the unseen instance via learning from the training examples associated with multiple class labels simultaneously [30, 46]. During the past decade, MLL has been widely applied to learn from the data containing rich semantics, such as multimedia content annotation [40, 33], text categorization [29, 27], music emotion analysis [21, 35], and bioinformatics analysis [3], etc.

However, in practice, obtaining ground-truth multiple labels for MLL training datasets is costly due to the expensive and time-consuming manual annotations. Comparing with multi-class learning where an example is associated with only *one positive label*, multi-label learning requires the *complete positive label set* for each example. On this account, the annotation cost of multi-label learning is significantly higher than multi-class classification, which limits its application especially when the number of categories is large.

To mitigate this problem, the setting of single-positive multi-label learning (SPMLL) [5] allows for significantly reduced annotations costs for the datasets, where each example is annotated with

---
*Corresponding author

36th Conference on Neural Information Processing Systems (NeurIPS 2022).

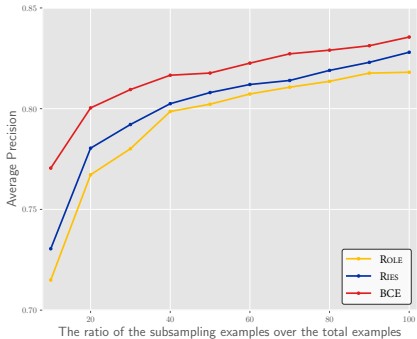

Figure 1: Test average precision on `tmc2007`. Each curve is generated by randomly subsampling the examples from the training set, where BCE is trained on fully labeled examples via binary cross-entropy loss while the SPMLL methods (ROLE [5] and the proposed SMILE) are trained on single-positive case.

only one relevant label. In Figure 1, comparing with fully labeled case, the SPMLL approaches on single-positive labeled examples only incur a tolerable drop in the performance but drastically reduce the amount of supervision required to train multi-label classifiers. By establishing the SPMLL methods with the learning power of DNNs, recent work [5] has also empirically validated that SPMLL would reduce the annotations costs while achieving good performance in practice. However, no method could provide theoretical insights as to why the model trained on the SPMLL examples can converge to an ideal one.

In this paper, we propose a theoretically-guaranteed method named SMILE, i.e., Single-positive MultI-label learning with Label Enhancement. Specifically, we first derive an unbiased risk estimator, which suggests that one positive label of each instance is sufficient to train the predictive models for multi-label learning. Besides, an estimation error bound is derived, which guarantees the risk-consistency [25] of the proposed method. Then we could design a benchmark solution via recovering the soft labels corresponding to each example in a label enhancement process [39, 36], where the posterior density of the latent soft labels is inferred by leveraging an approximate Beta density. The contributions are summarized as follows:

- Theoretically, we for the first time derive an unbiased risk estimator for SPMLL. Based on this, an estimation error bound is established that guarantees the risk-consistency and demonstrates that the obtained risk minimizer in SPMLL would approximately converge to the optimal risk minimizer in fully supervised MLL.
- Practically, we propose the method SMILE for SPMLL via adopting the latent soft labels recovered by label enhancement. The posterior density of the latent soft label is inferred by leveraging an approximate Beta density and the evidence lower bound (ELBO) [18] for the optimization is deduced.

Experiments on twelve corrupted MLL datasets show the effectiveness of SMILE over several existing SPMLL approaches.

## 2 Related Work

In multi-label learning, each example is associated with multiple class labels simultaneously. As the output space in MLL is exponential in size to the number of class labels, numerous approaches are proposed to exploit the label correlations to promote the learning process [14, 44, 31]. The first-order approaches disassemble the MLL problem into a number of binary classification problems [2, 45]. The second-order approaches consider the label correlations between pairs of labels [8, 11]. The high-order approaches further focus on the label correlations among the label set [26, 32]. Another line of research focuses on manipulating the feature space via formalizing label-specific feature to each class label to facilitate multi-label classification [23, 15, 42]. In addition, some work focus on dealing with MLL via deep models. A directed graph over the labels is established via employing

GCN to propagate information among all label nodes [4]. Transformer is leveraged for exploring the label dependency by introducing a ternary encoding scheme to represent the state of labels [20].

In practice, the labeling information is often incomplete in training data, since acquiring exhaustive supervision is extremely difficult. Numerous approaches have been proposed to handle the MLL with missing labels [12], which is also termed as MLL with partial labels [7]. A transductive learning method is proposed to concatenate features and labels and apply the matrix completion technique to it [12]. Then, the inductive learning method is proposed to exploit the structure of specific loss functions to offer efficient algorithms for learning with missing labels [41]. Wu [34] recovers the full labeling information of each training sample via enforcing consistency with available label-assignment and smoothness of label-assignment. The global and local label correlations are exploited simultaneously via learning a latent label representation in the missing label cases [47]. Durand [7] empirically compare different label-assignment strategies to show the potential to employ partial labels for MLL. Another method induces a cost function that measures the smoothness of labels and features to alleviate the overfitting issue when training data contains missing labels [16].

Comparing with multi-label learning with missing labels, SPMLL [5] considers the hardest version of this problem, where annotators are only asked to provide a single positive label for each training example and no additional negative or positive labels. When collecting multi-label annotations, it may be more efficient to annotate only one label rather than multiple labels for each example. To learn from SPMLL examples, an intuitive solution is "assume negative" (AN) [5], which assumes that unobserved labels are negative and trains the predictive model with binary cross-entropy loss on observed positive labels. Recent work [5] proposes some methods to reduce the damaging effects of false-negative labels. An expected positive regularization [5] is proposed to avoid the problem but the expected number of positive labels of each example should be given. Label smoothing [28] is employed to reduce the impact of the incorrect labels. Another approach [5] estimates the unobserved labels and encourages the classifier predictions to match the estimated labels via binary entropy loss. However, no methods can provide theoretical insights as to why the model trained on SPMLL examples can converge to an ideal one.

## 3 Problem Setup

### 3.1 Multi-Label Learning

In MLL, each example is associated with multiple labels, and aims to build a predictive model which can assign a set of relevant labels for the unseen instance. Let $\mathcal{X} = \mathbb{R}^q$ be the $q$-dimensional instance space and $\mathcal{Y} = \{1, 2, ..., c\}$ be the label space with $c$ class labels. Given the MLL training set $\mathcal{D} = \{(\boldsymbol{x}_i, Y_i) | 1 \leq i \leq n\}$ where $\boldsymbol{x}_i \in \mathcal{X}$ denotes the $q$-dimensional instance and $Y_i \in \mathcal{C}$ is the set of relevant labels associated with $\boldsymbol{x}_i$ where $\mathcal{C} = 2^{\mathcal{Y}}$. The task of multi-label learning is to induce a multi-label classifier $f : \mathcal{X} \mapsto 2^{\mathcal{Y}}$ that minimizes the following classification risk:

$$R(f) = \mathbb{E}_{p(\boldsymbol{x}, Y)} \left[ \mathcal{L} \left( f \left( \boldsymbol{x} \right), Y \right) \right]. \tag{1}$$

Here, $\mathcal{L} : \mathbb{R}^q \times 2^{\mathcal{Y}} \mapsto \mathbb{R}_+$ is a multi-label loss function that measures how well the model fits the data. Note that a method is risk-consistent if the method possesses a classification risk estimator that is equivalent to $R(f)$ given the same classifier $f$ [25, 9].

### 3.2 Single-Positive Multi-Label Learning

Given the SPMLL training set $\widetilde{\mathcal{D}} = \{(\boldsymbol{x}_i, \gamma_i) | 1 \leq i \leq n\}$ where $\gamma_i \in \mathcal{Y}$ denotes the observed single-positive label of $\boldsymbol{x}_i$. Note that $\gamma_i \in Y_i$ while its relevant label set $Y_i$ is not directly accessible to the learning algorithms. For each SPMLL training example $(\boldsymbol{x}_i, \gamma_i)$, we use the observed single-positive vector $\boldsymbol{l}_i = [l_i^1, l_i^2, \ldots, l_i^c]^\top \in \{0, 1\}^c$ to represent whether $j$-th label is the observed positive label, i.e., $l_i^j = 1$ if $j = \gamma_i$, otherwise $l_i^j = 0$. The multi-label vector is denoted by $\boldsymbol{y}_i = [y_i^1, y_i^2, \ldots, y_i^c]^\top \in \{0, 1\}^c$, where $y_i^j = 1$ if the $j$-th label is relevant to $\boldsymbol{x}_i$ and $y_i^j = 0$ if the $j$-th label is irrelevant. The task of SPMLL is to induce a multi-label classifier $f : \mathcal{X} \mapsto 2^{\mathcal{Y}}$ from $\widetilde{\mathcal{D}}$, which can assign a set of relevant label set for the unseen instance.

Recent work [5] empirically validates that SPMLL would reduce the amount of supervision with a tolerable damage in classification performance. The intuitive solution AN is assuming that unobserved

labels are negative, which leads to the drawback that introduces some number of false negative labels. Therefore, the SOTA approaches [5] aim to reduce the damaging effects of false-negative labels via employing the learning power of DNNs to achieve good performance in practice. However, there is no existing method that can provide theoretical insights.

## 4 The Proposed Method

### 4.1 Risk-Consistent Estimator

To deal with single-positive multi-label learning, the classification risk $R(f)$ in Eq. (1) could be rewritten as

$$
\begin{aligned}
&\mathbb{E}_{p(\boldsymbol{x},Y)}\left[\mathcal{L}\left(f\left(\boldsymbol{x}\right),Y\right)\right]\\
&=\int_{\boldsymbol{x}}\sum_{Y\in\mathcal{C}}\mathcal{L}\left(f\left(\boldsymbol{x}\right),Y\right)p(Y|\boldsymbol{x})p(\boldsymbol{x})\,\mathrm{d}\boldsymbol{x}\\
&=\int_{\boldsymbol{x}}\sum_{\gamma\in\mathcal{Y}}\sum_{Y\in\mathcal{C}}\mathcal{L}\left(f\left(\boldsymbol{x}\right),Y\right)\frac{p(Y|\boldsymbol{x})}{p(y^{\gamma}=1|\boldsymbol{x})c}p(y^{\gamma}=1|\boldsymbol{x})p(\boldsymbol{x})\,\mathrm{d}\boldsymbol{x}\\
&=\mathbb{E}_{p(\boldsymbol{x},\gamma)}\left[\frac{1}{p(y^{\gamma}=1|\boldsymbol{x})c}\sum_{Y\in\mathcal{C}}\mathcal{L}\left(f\left(\boldsymbol{x}\right),Y\right)p(Y|\boldsymbol{x})\right]\\
&=R_{sp}(f).
\end{aligned}
\tag{2}
$$

Additionally, we employ the widely used loss function in multi-label learning, i.e, binary cross-entropy loss, as the loss function $\mathcal{L}\left(f\left(\boldsymbol{x}\right),Y\right)$:

$$
\begin{aligned}
\mathcal{L}\left(f\left(\boldsymbol{x}\right),Y\right)&=\sum_{j\in Y}\log f_j(\boldsymbol{x})+\sum_{j\notin Y}\log\left(1-f_j(\boldsymbol{x})\right)\\
&=\sum_{j\in Y}\ell^j+\sum_{j\notin Y}\bar{\ell}^j,
\end{aligned}
\tag{3}
$$

where $\ell^j=\log f_j(\boldsymbol{x})$ and $\bar{\ell}^j=1-f_j(\boldsymbol{x})$. Then, $\sum_{Y\in\mathcal{C}}\mathcal{L}\left(f\left(\boldsymbol{x}\right),Y\right)p(Y|\boldsymbol{x})$ in Eq. (2) could be calculated as[1]

$$
\sum_{Y\in\mathcal{C}}\mathcal{L}\left(f\left(\boldsymbol{x}\right),Y\right)p(Y|\boldsymbol{x})=\sum_{j=1}^{c}d^j\ell^j+\left(1-d^j\right)\bar{\ell}^j.
\tag{4}
$$

Here, $d^j=p(y^j=1|\boldsymbol{x})\in[0,1]$ would be regarded as the soft label corresponding to class $j$ for $\boldsymbol{x}$. By substituting Eq. (4) into Eq. (2), we obtain the following risk-consistent estimator for SPMLL

$$
R_{sp}(f)=\mathbb{E}_{p(\boldsymbol{x},\gamma)}\left[\frac{1}{p(y^{\gamma}=1|\boldsymbol{x})c}\sum_{j=1}^{c}d^j\ell^j+(1-d^j)\bar{\ell}^j\right].
\tag{5}
$$

Therefore, we could express the empirical risk estimator via

$$
\widehat{R}_{sp}(f)=\frac{1}{n}\sum_{i=1}^{n}\left(\frac{1}{p(y^{\gamma_i}=1|\boldsymbol{x}_i)c}\sum_{j=1}^{c}d_i^j\ell_i^j+\left(1-d_i^j\right)\bar{\ell}_i^j\right).
\tag{6}
$$

Then, we could design a benchmark solution via applying the sigmoid function on $f_{\gamma_i}(\boldsymbol{x}_i)$ to approximate $p(y^{\gamma_i}=1|\boldsymbol{x}_i)$ and recovering the soft label $d_i^j$ corresponding to each example via the label enhancement process in the following subsection.

### 4.2 Training with Label Enhancement

To recover the soft label vector $\boldsymbol{d}_i=[d_i^1,d_i^2,\ldots,d_i^c]^\top\in[0,1]^c$, SMILE considers the topological information of the feature space and estimates adjacency matrix $\mathbf{A}=[a_{ij}]_{n\times n}$ with

$$
a_{ij}=\begin{cases}1 & \text{if }\boldsymbol{x}_i\in\mathcal{N}(\boldsymbol{x}_j)\\0 & \text{otherwise}\end{cases},
\tag{7}
$$

---

[1]The detail is provided in Appendix A.1.

where $\mathcal{N}(\boldsymbol{x}_j)$ is the set for $k$-nearest neighbors of $\boldsymbol{x}_j$.

We assume that the latent soft label matrix $\mathbf{D} = [\boldsymbol{d}_1, \boldsymbol{d}_2, \ldots, \boldsymbol{d}_n]$ generates the observed logical label matrix $\mathbf{L} = [\boldsymbol{l}_1, \boldsymbol{l}_2, \ldots, \boldsymbol{l}_n]$ and the adjacency matrix $\mathbf{A}$. Besides, the observed instance matrix $\mathbf{X} = [\boldsymbol{x}_1, \boldsymbol{x}_2, \ldots, \boldsymbol{x}_n]$ is generated from $\mathbf{D}$ and the latent feature matrix $\mathbf{Z} = [\boldsymbol{z}_1, \boldsymbol{z}_2, \ldots, \boldsymbol{z}_n]$. We assume that the prior density $p(\boldsymbol{d})$ is a Beta density with the minor values $\hat{\boldsymbol{\alpha}} = [\hat{\alpha}^1, \hat{\alpha}^2, \ldots, \hat{\alpha}^c]$ and $\hat{\boldsymbol{\beta}} = [\hat{\beta}^1, \hat{\beta}^2, \ldots, \hat{\beta}^c]$, i.e., $p(\boldsymbol{d}) = \prod_{j=1}^{c} \text{Beta}\left(d^j \mid \hat{\alpha}^j, \hat{\beta}^j\right)$. Then the prior density $p(\mathbf{D})$ could be the product of each $p(\boldsymbol{d})$. In addition, We assume that the prior density $p(\boldsymbol{z})$ is a standard Gaussian and prior density $p(\mathbf{Z})$ can be represented as the product of each Gaussian $p(\mathbf{Z}) = \prod_{i=1}^{n} Gau(\boldsymbol{z}_i|\boldsymbol{0}, \boldsymbol{1})$. Then, the posterior density $p(\mathbf{D}, \mathbf{Z}|\mathbf{L}, \mathbf{X}, \mathbf{A})$ can be decomposed as follows:

$$p(\mathbf{D}, \mathbf{Z}|\mathbf{L}, \mathbf{X}, \mathbf{A}) = p(\mathbf{D}|\mathbf{L}, \mathbf{X}, \mathbf{A})p(\mathbf{Z}|\mathbf{D}, \mathbf{L}, \mathbf{X}, \mathbf{A}) = p(\mathbf{D}|\mathbf{L}, \mathbf{X}, \mathbf{A})p(\mathbf{Z}|\mathbf{D}, \mathbf{X}) \qquad (8)$$

where $\mathbf{L}$ and $\mathbf{A}$ can be removed from the condition of $p(\mathbf{Z}|\mathbf{D}, \mathbf{L}, \mathbf{X}, \mathbf{A})$ because of the independence between $\mathbf{Z}$ and $\mathbf{L}, \mathbf{A}$ when latent variable $\mathbf{D}$ is given in the condition. Here we employ $q(\mathbf{D}|\mathbf{L}, \mathbf{X}, \mathbf{A})$ and $q(\mathbf{Z}|\mathbf{D}, \mathbf{X})$ to approximate the true posterior $p(\mathbf{D}|\mathbf{L}, \mathbf{X}, \mathbf{A})$ and $p(\mathbf{Z}|\mathbf{D}, \mathbf{X})$ respectively. The approximate posterior $q(\mathbf{D}|\mathbf{L}, \mathbf{X}, \mathbf{A})$ could be the product of Beta parameterized by $\boldsymbol{\alpha}_i = [\alpha_i^1, \alpha_i^2, \ldots, \alpha_i^c]^\top$ and $\boldsymbol{\beta}_i = [\beta_i^1, \beta_i^2, \ldots, \beta_i^c]^\top$:

$$q_{\boldsymbol{w}_1}(\mathbf{D} \mid \mathbf{L}, \mathbf{X}, \mathbf{A}) = \prod_{i=1}^{n} \prod_{j=1}^{c} \text{Beta}\left(d_i^j|\alpha_i^j, \beta_i^j\right). \qquad (9)$$

Here, the parameters $\boldsymbol{\Delta} = [\boldsymbol{\alpha}_1, \boldsymbol{\alpha}_2, \ldots, \boldsymbol{\alpha}_n]$ and $\boldsymbol{\Phi} = [\boldsymbol{\beta}_1, \boldsymbol{\beta}_2, \ldots, \boldsymbol{\beta}_n]$ are the outputs of the inference model parameterized by $\boldsymbol{w}_1$ as a GCN [19] with adjacency matrix by $\mathbf{A}$. Let $q_{\boldsymbol{w}_2}(\mathbf{Z}|\mathbf{D}, \mathbf{X})$ be the product of Gaussian parameterized by the mean vector $\boldsymbol{\mu}_i$ and standard deviation vector $\boldsymbol{\sigma}_i$:

$$q_{\boldsymbol{w}_2}(\mathbf{Z}|\mathbf{D}, \mathbf{X}) = \prod_{i=1}^{n} Gau(\boldsymbol{z}_i|\boldsymbol{\mu}_i, \boldsymbol{\sigma}_i), \qquad (10)$$

The parameters $\boldsymbol{\Lambda} = [\boldsymbol{\mu}_1, \boldsymbol{\mu}_2, \ldots, \boldsymbol{\mu}_n, \boldsymbol{\sigma}_1, \boldsymbol{\sigma}_2, \ldots, \boldsymbol{\sigma}_n]$ are the outputs of the inference model with a MLP parameterized by $\boldsymbol{w}_2$.

We derive the evidence lower bound (ELBO) [18] on the marginal likelihood of the model to ensure that $q_{\boldsymbol{w}}(\mathbf{D}, \mathbf{Z}|\mathbf{L}, \mathbf{X}, \mathbf{A})$ is as close as possible to $p(\mathbf{D}, \mathbf{Z}|\mathbf{L}, \mathbf{X}, \mathbf{A})$ [1]:

$$\begin{aligned}
\mathcal{L}_{ELBO} = &\mathbb{E}_{q_{\boldsymbol{w}}(\mathbf{D},\mathbf{Z}|\mathbf{L},\mathbf{X},\mathbf{A})}[\log p(\mathbf{X}|\mathbf{D}, \mathbf{Z}) + \log p(\mathbf{L}|\mathbf{D}) + \log p(\mathbf{A}|\mathbf{D})] \\
&- \text{KL}[q_{\boldsymbol{w}_1}(\mathbf{D}|\mathbf{L}, \mathbf{X}, \mathbf{A})||p(\mathbf{D})] - \text{KL}[q_{\boldsymbol{w}_2}(\mathbf{Z}|\mathbf{D}, \mathbf{X})||p(\mathbf{Z})].
\end{aligned} \qquad (11)$$

We further assume that $p(\mathbf{X}|\mathbf{D}, \mathbf{Z})$ is a product of each Gaussian with means $\boldsymbol{\xi}_i$ and $p(\mathbf{L}|\mathbf{D})$ is a product of each multivariate Bernoulli with probabilities $\boldsymbol{\tau}_i$. In order to simplify the observation model, $\mathbf{T}^{(m)} = [\boldsymbol{\tau}_1^{(m)}, \boldsymbol{\tau}_2^{(m)}, \ldots, \boldsymbol{\tau}_n^{(m)}]$ is computed from $m$-th sampling $\mathbf{D}^{(m)}$ with a MLP parameterized by $\boldsymbol{\eta}_1$ and $\boldsymbol{\Xi}^{(m)} = [\boldsymbol{\xi}_1^{(m)}, \boldsymbol{\xi}_2^{(m)}, \ldots, \boldsymbol{\xi}_n^{(m)}]$ is computed from $m$-th sampling $\mathbf{D}^{(m)}$ and $\mathbf{Z}^{(m)}$ with a MLP parameterized by $\boldsymbol{\eta}_2$. Then the first part of Eq. (11) can be tractable as

$$\begin{aligned}
&\mathbb{E}_{q_{\boldsymbol{w}}(\mathbf{D},\mathbf{Z}|\mathbf{L},\mathbf{X},\mathbf{A})}[\log p(\mathbf{X}|\mathbf{D}, \mathbf{Z}) + \log p(\mathbf{L}|\mathbf{D}) + \log p(\mathbf{A}|\mathbf{D})] = \frac{1}{M} \sum_{m=1}^{M} \text{tr}\left(\mathbf{L}^\top \log \mathbf{T}^{(m)}\right) \\
&+ \text{tr}\left((\mathbf{I} - \mathbf{L})^\top \log\left(\mathbf{I} - \mathbf{T}^{(m)}\right)\right) - \|\mathbf{A} - S\left(\mathbf{D}^{(m)}\mathbf{D}^{(m)\top}\right)\|_F^2 + \|\boldsymbol{\Xi}^{(m)} - \mathbf{X}\|_F^2
\end{aligned} \qquad (12)$$

Here, $S(\cdot)$ is the logistic sigmoid function, and implicit reparameterization trick [10] and MC sampling [18, 37, 38] are employed.

The second part of Eq. (11) can be analytically calculated as

$$\begin{aligned}
\text{KL}\left(q_{\boldsymbol{w}_1}(\mathbf{D}|\mathbf{L}, \mathbf{X}, \mathbf{A})||p(\mathbf{D})\right) = &\sum_{i=1}^{n} \sum_{j=1}^{c} \log \frac{\Gamma(\alpha_i^j + \beta_i^j)\Gamma(\hat{\alpha}_i^j)\Gamma(\hat{\beta}_i^j)}{\Gamma(\hat{\alpha}_i^j + \hat{\beta}_i^j)\Gamma(\alpha_i^j)\Gamma(\beta_i^j)} + (\alpha_i^j - \hat{\alpha}_i^j)\psi(\alpha_i^j) \\
&- (\alpha_i^j - \hat{\alpha}_i^j + \beta_i^j - \hat{\beta}_i^j)\psi(\alpha_i^j + \beta_i^j) + (\beta_i^j - \hat{\beta}_i^j)\psi(\beta_i^j).
\end{aligned} \qquad (13)$$

---

[1]The detail is provided in Appendix A.2.

**Algorithm 1** SMILE Algorithm

---

**Input:** The SPMLL training set $\widetilde{\mathcal{D}} = \{(\boldsymbol{x}_i, \gamma_i)\}_{i=1}^n$, the number of iteration $I$ and the number of epoch $T$;
1: Warm-up $\boldsymbol{\theta}$ by using AN solution, and initialize the reference model $\boldsymbol{w}_1$, $\boldsymbol{w}_2$ and observation model $\boldsymbol{\eta}$;
2: Estimate the adjacency matrix $\mathbf{A}$ by Eq. (7);
3: **for** $t = 1, \ldots, T$ **do**
4:     Shuffle training set $\widetilde{\mathcal{D}} = \{(\boldsymbol{x}_i, \gamma_i)\}_{i=1}^n$ into $I$ mini-batches;
5:     **for** $k = 1, \ldots, I$ **do**
6:         Update $\boldsymbol{w}_1$, $\boldsymbol{w}_2$ and $\boldsymbol{\eta}$ by forward computation and back-propagation by Eq. (16);
7:         Obtain the soft label $\boldsymbol{d}_i$ for each example $\boldsymbol{x}_i$ by Eq. (9);
8:         Apply the sigmoid function on $f_{\gamma_i}(\boldsymbol{x}_i)$ to approximate $p(y^{\gamma_i} = 1|\boldsymbol{x}_i)$;
9:         Update $\boldsymbol{\theta}$ by forward computation and back-propagation by Eq. (6);
10:    **end for**
11: **end for**
**Output:** The predictive model $\boldsymbol{\theta}$.

---

Here, $\Gamma(\cdot)$ and $\psi(\cdot)$ are Gamma function and Digamma function, respectively. The third part of Eq. (11) can be analytically calculated as follows:

$$\text{KL}(q_{\boldsymbol{w}_2}(\mathbf{Z}|\mathbf{D}, \mathbf{X})||p(\mathbf{Z})) = \sum_{i=1}^n \sum_{j=1}^J \left( 1 + \log\left((\sigma_i^j)\right) - (\mu_i^j)^2 - (\sigma_i^j)^2 \right). \tag{14}$$

Besides, we could promote the label enhancement process via enforcing that the estimated $\mathbf{D}$ should inherit the labeling-information of observed labels:

$$T_C = -\frac{1}{n} \sum_{i=1}^n \sum_{j=1}^c l_i^j \log d_i^j + \left(1 - l_i^j\right)\left(1 - \log d_i^j\right). \tag{15}$$

Finally, the objective of label enhancement $T_{LE}$ is obtained:

$$T_{LE} = -\lambda \mathcal{L}_{ELBO} + T_C, \tag{16}$$

where $\lambda$ is a hyper-parameter.

SMILE first initializes the predictive network by warm-up training with AN solution, which would attain a fine network before it starts fitting noise [43]. Then we could sample the soft label from fixed Beta after label enhancement and the sigmoid function on $f_{\gamma_i}(\boldsymbol{x}_i)$ to approximate $p(y^{\gamma_i} = 1|\boldsymbol{x}_i)$ to make Eq. (6) accessible, and train the predictive model $\boldsymbol{\theta}$ by minimizing the risk estimator. In each epoch, SMILE alternately operates label enhancement process and classifier training process. Algorithm 1 shows the algorithmic description of SMILE.

### 4.3 Estimation Error Bound

In this subsection, we establish an estimation error bound of the proposed method. The empirical risk estimator according to Eq.(6) can be rewritten as:

$$\widehat{R}_{sp}(f) = \frac{1}{n} \sum_{i=1}^n \sum_{j=1}^L \left( w_i^j \ell_i^j + \bar{w}_i^j \bar{\ell}_i^j \right), \tag{17}$$

where $w_i^j = \frac{d_i^j}{p(y^\gamma = 1|\boldsymbol{x}_i)c}$ and $\bar{w}_i^j = \frac{1 - d_i^j}{p(y^\gamma = 1|\boldsymbol{x}_i)c}$. Then the loss function $\mathcal{L}_{sp}$ is

$$\mathcal{L}_{sp} = \sum_{j=1}^L \left( w_i^j \ell_i^j + \bar{w}_i^j \bar{\ell}_i^j \right). \tag{18}$$

Table 1: Predictive performance of each comparing approach (mean±std) in terms of *Average precision* ↑. The best performance (the larger the better) is shown in bold face.

| Datasets | SMILE | AN | AN-LS | WAN | ROLE | GLOCAL | MLML | D2ML |
|---|---|---|---|---|---|---|---|---|
| CAL500 | **0.401±0.011** | 0.382±0.044 | 0.253±0.031 | 0.393±0.011 | 0.288±0.008 | 0.227±0.002 | 0.233±0.000 | 0.223±0.001 |
| image | **0.784±0.044** | 0.613±0.081 | 0.621±0.073 | 0.685±0.058 | 0.696±0.039 | 0.771±0.003 | 0.652±0.001 | 0.274±0.003 |
| scene | **0.841±0.070** | 0.740±0.127 | 0.741±0.117 | 0.801±0.020 | 0.717±0.067 | 0.825±0.001 | 0.814±0.000 | 0.285±0.002 |
| yeast | **0.758±0.003** | 0.755±0.003 | 0.753±0.003 | 0.757±0.003 | 0.753±0.003 | 0.646±0.002 | 0.456±0.002 | 0.323±0.001 |
| corel5k | **0.303±0.007** | 0.299±0.005 | 0.272±0.005 | 0.302±0.004 | 0.215±0.011 | 0.218±0.001 | 0.072±0.001 | 0.028±0.001 |
| rcv1-s1 | **0.616±0.001** | 0.604±0.004 | 0.581±0.002 | 0.610±0.005 | 0.570±0.004 | 0.229±0.000 | 0.221±0.003 | 0.053±0.001 |
| corel16k-s1 | **0.344±0.003** | 0.337±0.003 | 0.316±0.002 | 0.344±0.003 | 0.288±0.004 | 0.029±0.001 | 0.081±0.001 | 0.029±0.004 |
| delicious | 0.319±0.001 | 0.297±0.009 | 0.193±0.005 | **0.320±0.001** | 0.199±0.004 | 0.027±0.001 | 0.086±0.001 | 0.028±0.001 |
| iaprtc12 | **0.314±0.003** | 0.292±0.008 | 0.244±0.008 | 0.266±0.006 | 0.243±0.005 | 0.035±0.002 | 0.126±0.001 | 0.026±0.001 |
| espgame | **0.259±0.003** | 0.248±0.002 | 0.208±0.003 | 0.259±0.002 | 0.216±0.004 | 0.038±0.000 | 0.086±0.002 | 0.038±0.001 |
| mirflickr | **0.635±0.004** | 0.629±0.003 | 0.604±0.004 | 0.611±0.004 | 0.545±0.019 | 0.476±0.000 | 0.253±0.003 | 0.132±0.002 |
| tmc2007 | **0.820±0.002** | 0.815±0.003 | 0.802±0.003 | 0.815±0.001 | 0.798±0.005 | 0.649±0.000 | 0.415±0.000 | 0.161±0.001 |

We define a function space as:

$$\mathcal{G}_{sp} = \left\{ (\boldsymbol{x}, y) \mapsto \sum_{j=1}^{L} \left( w^j \ell^j + \bar{w}^j \bar{\ell}^j \right) | f \in \mathcal{F} \right\}, \tag{19}$$

and denote the expected Rademacher complexity [1] of $\mathcal{G}_{sp}$ as:

$$\widetilde{\mathfrak{R}}_n(\mathcal{G}_{sp}) = \mathbb{E}_{\boldsymbol{x}, y, \boldsymbol{\sigma}} \left[ \sup_{g \in \mathcal{G}_{sp}} \frac{1}{n} \sum_{i=1}^{n} \sigma_i g\left(\boldsymbol{x}_i, y_i\right) \right], \tag{20}$$

where $\boldsymbol{\sigma} = \{\sigma_1, \sigma_2, \ldots, \sigma_n\}$ is $n$ Rademacher variables with $\sigma_i$ independently uniform variable taking value in $\{+1, -1\}$. Then we have

**Lemma 1** *We suppose that the SPMLL loss function $\mathcal{L}_{sp}$ could be bounded by $M$, i.e., $M = \sup_{\boldsymbol{x} \in \mathcal{X}, f \in \mathcal{F}, y \in \mathcal{Y}} \mathcal{L}_{sp}\left(f(\boldsymbol{x}), y\right)$, and for any $\delta > 0$, with probability at least $1 - \delta$, then we have*

$$\sup_{f \in \mathcal{F}} \left| R_{sp}(f) - \widehat{R}_{sp}(f) \right| \leq 2\widetilde{\mathfrak{R}}_n(\mathcal{G}_{sp}) + \frac{M}{2} \sqrt{\frac{\log \frac{2}{\delta}}{2n}}.$$

The proof of Lemma 1 could be founded in Appendix A.3.

**Lemma 2** *We suppose that the loss function $\ell\left(f(\boldsymbol{x}), y\right)$ and $\bar{\ell}\left(f(\boldsymbol{x}), y\right)$ are $\rho^+$-Lipschitz and $\rho^-$-Lipschitz with respect to $f(\boldsymbol{x})$ $(0 < \rho^+ < \infty$ and $0 < \rho^- < \infty)$ for all $y \in \mathcal{Y}$, respectively, and $w^j$ and $\bar{w}^j$ are both bounded in $[0, \kappa]$. Then, we have*

$$\widetilde{\mathfrak{R}}_n(\mathcal{G}_{sp}) \leq \sqrt{2}\kappa c(\rho^+ + \rho^-) \sum_{j=1}^{c} \mathfrak{R}_n(\mathcal{H}_{y_j}),$$

*where $\mathcal{H}_y = \{h : \boldsymbol{x} \mapsto f_y(\boldsymbol{x}) | f \in \mathcal{F}\}$ and $\mathfrak{R}_n(\mathcal{H}_y) = \mathbb{E}_{\boldsymbol{x}, \boldsymbol{\sigma}} \left[\sup_{h \in \mathcal{H}_y} \frac{1}{n} \sum_{i=1}^{n} h\left(\boldsymbol{x}_i\right)\right]$.*

The proof of Lemma 2 could be founded in Appendix A.4.

Based one Lemma 1 and 2, we could obtain the following theorem:

**Theorem 1** *Assume the loss function $\ell\left(f(\boldsymbol{x}), y\right)$ and $\bar{\ell}\left(f(\boldsymbol{x}), y\right)$ are $\rho^+$-Lipschitz and $\rho^-$-Lipschitz with respect to $f(\boldsymbol{x})$ $(0 < \rho^+ < \infty$ and $0 < \rho^- < \infty)$ for all $y \in \mathcal{Y}$ and the loss function $\mathcal{L}_{sp}$ are bounded by $M$, i.e., $M = \sup_{\boldsymbol{x} \in \mathcal{X}, f \in \mathcal{F}, y \in \mathcal{Y}} \mathcal{L}_{sp}\left(f(\boldsymbol{x}), y\right)$, with probability at least $1 - \delta$,*

$$R(\widehat{f}_{sp}) - R(f^*) \leq 4\sqrt{2}\kappa c(\rho^+ + \rho^-) \sum_{j=1}^{c} \mathfrak{R}_n(\mathcal{H}_y) + M \sqrt{\frac{\log \frac{2}{\delta}}{2n}}.$$

Here, $\widehat{f}_{sp} = \min_{f \in \mathcal{F}} \widehat{R}_{sp}(f)$ and $f^\star = \min_{f \in \mathcal{F}} R(f)$ are the empirical risk minimizer and the true risk minimizer, respectively. The proof could be founded in Appendix A.5. Theorem 1 shows that $f_{sp}$ would converge to $f^\star$ as $n \to \infty$ and $\mathfrak{R}_n\left(\mathcal{H}_y\right) \to 0$.

Table 2: Predictive performance of each comparing approach (mean±std) in terms of *One-error* ↓. The best performance (the smaller the better) is shown in bold face.

| Datasets | SMILE | AN | AN-LS | WAN | ROLE | GLOCAL | MLML | D2ML |
|---|---|---|---|---|---|---|---|---|
| CAL500 | 0.358±0.156 | **0.325±0.134** | 0.627±0.188 | 0.420±0.152 | 0.557±0.034 | 0.843±0.011 | 0.839±0.032 | 0.833±0.003 |
| image | 0.350±0.046 | 0.597±0.102 | 0.577±0.095 | 0.516±0.090 | 0.488±0.055 | 0.365±0.012 | **0.200±0.023** | 0.600±0.019 |
| scene | 0.278±0.112 | 0.417±0.152 | 0.412±0.145 | 0.344±0.033 | 0.477±0.111 | 0.286±0.024 | **0.167±0.011** | 0.667±0.023 |
| yeast | **0.236±0.008** | 0.242±0.012 | 0.244±0.009 | 0.242±0.011 | 0.239±0.010 | 0.276±0.032 | 0.285±0.003 | 0.500±0.022 |
| corel5k | **0.648±0.008** | 0.685±0.019 | 0.674±0.013 | 0.656±0.013 | 0.696±0.022 | 0.764±0.011 | 0.947±0.005 | 0.987±0.003 |
| rcv1-s1 | **0.438±0.007** | 0.445±0.011 | 0.467±0.008 | 0.449±0.011 | 0.464±0.003 | 0.810±0.029 | 0.782±0.002 | 0.941±0.004 |
| corel16k-s1 | **0.641±0.008** | 0.655±0.005 | 0.666±0.007 | 0.642±0.009 | 0.667±0.009 | 0.989±0.003 | 0.830±0.001 | 0.987±0.003 |
| delicious | 0.410±0.005 | 0.454±0.026 | 0.516±0.021 | **0.405±0.007** | 0.498±0.012 | 0.996±0.003 | 0.804±0.011 | 0.967±0.003 |
| iaprtc12 | **0.579±0.022** | 0.604±0.022 | 0.618±0.017 | 0.662±0.015 | 0.659±0.016 | 0.997±0.000 | 0.605±0.012 | 0.897±0.006 |
| espgame | **0.662±0.007** | 0.686±0.012 | 0.702±0.012 | 0.673±0.008 | 0.707±0.009 | 0.995±0.000 | 0.699±0.003 | 0.734±0.004 |
| mirflickr | **0.335±0.013** | 0.343±0.017 | 0.343±0.006 | 0.385±0.009 | 0.497±0.030 | 0.670±0.054 | 0.447±0.011 | 0.816±0.007 |
| tmc2007 | **0.204±0.003** | 0.215±0.003 | 0.221±0.006 | 0.220±0.003 | 0.225±0.004 | 0.313±0.001 | 0.227±0.002 | 0.409±0.008 |

Table 3: Summary of the Wilcoxon signed-ranks test for SMILE against other comparing approaches at 0.05 significance level. The *p*-values are shown in the brackets.

| SMILE against | AN | AN-LS | WAN | ROLE | GLOCAL | MLML | D2ML |
|---|---|---|---|---|---|---|---|
| *Average precision* | **win**[0.0005] | **win**[0.0005] | **win**[0.0092] | **win**[0.0005] | **win**[0.0005] | **win**[0.0005] | **win**[0.0005] |
| *One-error* | **win**[0.0122] | **win**[0.0005] | **win**[0.0015] | **win**[0.0005] | **win**[0.0005] | **win**[0.0342] | **win**[0.0005] |
| *Ranking loss* | **win**[0.0269] | **win**[0.0005] | **tie**[0.1533] | **win**[0.0005] | **win**[0.0005] | **win**[0.0024] | **win**[0.0005] |
| *Hamming loss* | **win**[0.0277] | **win**[0.0178] | **win**[0.0005] | **win**[0.0277] | **win**[0.0277] | **win**[0.0277] | **win**[0.0077] |
| *Coverage* | **win**[0.0425] | **win**[0.0005] | **tie**[0.1819] | **win**[0.0005] | **win**[0.0005] | **win**[0.0024] | **win**[0.0015] |

## 5 Experiments

### 5.1 Experimental Configurations

In the experiments, we adopt twelve widely-used MLL datasets [13], which cover a broad range of cases with diversified multi-label properties. To evaluate the performance of SPMLL methods, we generate the single positive training data by randomly selecting one positive label to keep for each training example in the MLL datasets. For each dataset, we run the comparing methods with 80%/10%/10% train/validation/test split. The validation and test sets are always fully labeled. The detailed descriptions of these datasets are provided in Appendix A.6. Five popular multi-label metrics *Ranking loss*, *Hamming loss*, *One-error*, *Coverage*, and *Average precision* [46] are employed for performance evaluation. Furthermore, for *Average precision*, the *larger* the values the better the performance. While for the other four metrics, the *smaller* the values the better the performance.

In this paper, SMILE is compared against four well-established SPMLL approaches including 1) AN [5] which assumes unobserved labels are negative and employs binary entropy loss for the training examples with the modified labels, 2) AN-LS [5] which assumes unobserved labels are negative and employs label smoothing [28] to reduce the impact of the incorrect labels (i.e. those labels incorrectly assumed to be negative), 3) WAN [5] which reduces the impact of false negatives by employing the down-weight terms in the loss corresponding to negative labels, and 4) ROLE [5] which online estimates of the unobserved labels throughout training and encourages the classifier predictions to match the estimated labels via binary entropy loss.

As SPMLL could be regarded as the hardest version of MLL with missing labels, three well-established approaches of MLL with missing labels (also termed as MLL with partial labels) are adopted as the comparing approaches including 1) GLOCAL [47] which exploits global and local label correlations simultaneously via learning a latent label representation in the missing label cases, 2) MLML [34] which recovers the full label assignment for each sample by enforcing consistency with available label assignments and smoothness of labels, and 3) D2ML [24] which utilizes both local low-rank label structures and label discriminant information for learning from missing labels.

For all the DNN-based approaches (AN, AN-LS, WAN, ROLE and SMILE), we adopt three-layer MLP as the predictive model for fair comparisons and use the Adam optimizer [17]. The mini-batch size and the number of epochs are set to 16 and 25, respectively. The learning rate and weight decay are selected from $\{10^{-4}, 10^{-3}, 10^{-2}\}$ with a validation set. Other hyper-parameters for all the comparing methods are also selected based on the validation. All the comparing methods run 5 trials (with 80%/10%/10% train/validation/test split) on each dataset.

Table 4: Predictive performance of SMILE and its variant (mean±std) in terms of *Average precision*, *One-error*, and *Ranking loss*.

| Datasets | Average precision↑ | | One-error↓ | | Ranking loss↓ | |
|---|---|---|---|---|---|---|
| | SMILE | SMILE-SI | SMILE | SMILE-SI | SMILE | SMILE-SI |
| CAL500 | 0.401±0.011 | **0.409±0.000** | 0.358±0.156 | **0.343±0.010** | **0.239±0.010** | 0.244±0.000 |
| image | **0.784±0.044** | 0.547±0.022 | **0.350±0.046** | 0.675±0.035 | **0.170±0.055** | 0.428±0.005 |
| scene | **0.841±0.070** | 0.711±0.057 | **0.278±0.112** | 0.483±0.093 | **0.086±0.045** | 0.168±0.037 |
| yeast | **0.758±0.003** | 0.738±0.002 | **0.236±0.008** | 0.246±0.006 | **0.161±0.003** | 0.176±0.001 |
| corel5k | **0.303±0.007** | 0.302±0.003 | **0.648±0.008** | 0.655±0.007 | 0.134±0.003 | **0.116±0.000** |
| rcv1-s1 | **0.616±0.001** | 0.577±0.004 | **0.438±0.007** | 0.477±0.015 | **0.042±0.000** | 0.055±0.000 |
| corel16k-s1 | **0.344±0.003** | 0.336±0.001 | **0.641±0.008** | 0.649±0.002 | **0.133±0.001** | 0.140±0.000 |
| delicious | **0.319±0.001** | 0.293±0.004 | **0.410±0.005** | 0.434±0.023 | **0.126±0.000** | 0.146±0.002 |
| iaprtc12 | **0.314±0.003** | 0.287±0.001 | **0.579±0.022** | 0.607±0.015 | **0.115±0.002** | 0.143±0.002 |
| espgame | **0.259±0.003** | 0.249±0.003 | **0.662±0.007** | 0.675±0.002 | **0.158±0.001** | 0.170±0.002 |
| mirflickr | **0.635±0.004** | 0.628±0.001 | **0.335±0.013** | 0.349±0.012 | **0.117±0.002** | 0.122±0.002 |
| tmc2007 | **0.820±0.002** | 0.814±0.001 | **0.204±0.003** | 0.210±0.003 | 0.049±0.001 | **0.048±0.000** |

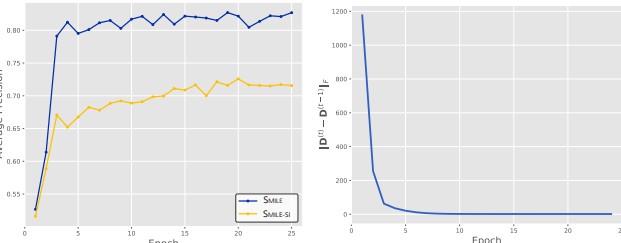

Figure 2: *Average precision*. Figure 3: Convergence of **D**.

Table 5: Wilcoxon signed ranks test(at 0.05 significance level).

| Evaluation metric | SMILE against SMILE-SI | |
|---|---|---|
| | performance | $p$-value |
| *Average precision* | **win** | 0.0049 |
| *One-error* | **win** | 0.0092 |
| *Ranking loss* | **win** | 0.0161 |
| *Hamming loss* | **win** | 0.0277 |
| *Coverage* | **win** | 0.0122 |

## 5.2 Experimental Results

Tables 1 and 2 report the results of all approaches on *Average precision* and *One-error*, respectively. For each evaluation metric, "↑" indicates "the smaller the better" while "↓" indicates "the larger the better". The results on other metrics are similar and could be seen in Appendix A.6. In addition, Wilcoxon signed-ranks test [6] is employed to show whether SMILE has a significant performance than other comparing approaches. Table 3 reports the $p$-values for the corresponding tests and the statistical test results at 0.05 significance level.

Table 3 shows that SMILE achieves superior performance against all the comparing approaches on all evaluation metrics (except on *Rranking loss* and *Coverage* where SMILE achieves comparable performance against WAN). The superior performance of SMILE provides a strong evidence for the effectiveness of risk-consistent estimator for SPMLL. Tables 1 and 2 show that the performance advantage of SMILE over comparing approaches is stable under varying the number of class labels. In summary, these experimental results clearly validate the effectiveness of SMILE.

## 5.3 Further Analysis

To show the helpfulness of label enhancement to SMILE, a vanilla variant of SMILE (named SMILE-SI) is adopted. Here, label enhancement is replaced by approximating $d_i^j$ with the confidence of the current model $f_j(\boldsymbol{x}_i)$, which is a widely-used technique [9, 22] to approximate the soft label in weakly supervised learning. Table 4 reports detailed experimental results in terms of *Average precision*, *One-error*, and *Ranking loss*, respectively. The detailed experimental results in terms of other metrics are reported in Appendix A.6. Besides, the performance of each approach with the number of epochs on scene is shown in Figure 2. Wilcoxon signed-ranks test [6] in Table 5 shows that SMILE achieves superior performance against SMILE-SI on all evaluation metrics, which clearly validates the usefulness of label enhancement. Figure 3 illustrates the estimated **D** converges with the number of epochs on delicious, which shows that the estimated soft label could converge efficiently.

# 6 Conclusion

In this paper, we study single-positive multi-label learning and propose a novel approach SMILE. We derive an unbiased risk estimator, which suggests that one positive label of each instance is sufficient to train predictive models for multi-label learning, and design a benchmark solution via estimating the soft label corresponding to each example in a label enhancement process. The effectiveness of the proposed method is validated on twelve corrupted MLL datasets.

# 7 Acknowledgments

This research was supported by the National Key Research & Development Plan of China (No. 2021ZD0114202), the National Science Foundation of China (62206050, 62125602, and 62076063), China Postdoctoral Science Foundation (2021M700023), Jiangsu Province Science Foundation for Youths (BK20210220).

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
