# One Positive Label is Sufficient:
# Single-Positive Multi-Label Learning with Label Enhancement

**Ning Xu[1], Congyu Qiao[1], Jiaqi Lv[2], Xin Geng[1],\*Min-Ling Zhang[1]**
[1]School of Computer Science and Engineering, Southeast University, Nanjing 210096, China
[2] RIKEN Center for Advanced Intelligence Project, Tokyo 103-0027, Japan
{xning, qiaocy}@seu.edu.cn, is.jiaqi.lv@gmail.com,
{xgeng, zhangml}@seu.edu.cn

## A  Appendix

### A.1  Calculation Details of Eq. (4)

$$
\begin{aligned}
&\sum_{Y \in \mathcal{C}} \mathcal{L}\left(f\left(\boldsymbol{x}\right), Y\right) p(Y|\boldsymbol{x}) \\
&= \sum_{Y \in \mathcal{C}} \sum_{j \in Y} \ell^j p(Y|\boldsymbol{x}) + \sum_{Y \in \mathcal{C}} \sum_{j \notin Y} \bar{\ell}^j p(Y|\boldsymbol{x}) \\
&= \sum_{j=1}^{c} \ell^j \sum_{Y \in \mathcal{C}_j} p(Y|\boldsymbol{x}) + \sum_{j=1}^{c} \bar{\ell}^j \sum_{Y \in \widetilde{\mathcal{C}}_j} p(Y|\boldsymbol{x}) \\
&= \sum_{j=1}^{c} p(y^j = 1|\boldsymbol{x})\ell^j \sum_{Y \in \mathcal{C}_j} \prod_{k \in Y, k \neq j} p(y^k = 1|\boldsymbol{x}) \prod_{k \notin Y}\left(1 - p(y^k = 1|\boldsymbol{x})\right) + \\
&\quad \sum_{j=1}^{c}\left(1 - p(y^j = 1|\boldsymbol{x})\right) \bar{\ell}^j \sum_{Y \in \widetilde{\mathcal{C}}_j} \prod_{k \in Y} p(y^k = 1|\boldsymbol{x}) \prod_{k \notin Y, k \neq j}\left(1 - p(y^k = 1|\boldsymbol{x})\right) \\
&= \sum_{j=1}^{c} p(y^j = 1|\boldsymbol{x})\ell^j + \left(1 - p(y^j = 1|\boldsymbol{x})\right) \bar{\ell}^j \\
&= \sum_{j=1}^{c} d^j \ell^j + \left(1 - d^j\right) \bar{\ell}^j.
\end{aligned}
\tag{1}
$$

where $d^j = p(y^j = 1|\boldsymbol{x})$, $\mathcal{C}_j$ denotes the subset of $\mathcal{C}$ which contains label $j$ and $\widetilde{\mathcal{C}}_j$ denotes the subset of $\mathcal{C}$ without label $j$.

### A.2  Calculation Details of Eq. (4)

$$
\log p(\mathbf{L}, \mathbf{X}, \mathbf{A}) = \log \mathbf{p}(\mathbf{D}, \mathbf{Z}, \mathbf{L}, \mathbf{X}, \mathbf{A}) - \log \mathbf{p}(\mathbf{D}, \mathbf{Z}|\mathbf{L}, \mathbf{X}, \mathbf{A}).
\tag{2}
$$

---

\*Corresponding author

36th Conference on Neural Information Processing Systems (NeurIPS 2022).

Multiply both sides by $q_{\mathbf{w}}(\mathbf{Z}, \mathbf{D}|\mathbf{L}, \mathbf{X}, \mathbf{A})$, and for $\mathbf{D}$ and $\mathbf{Z}$ integral:

$$
\int_{\mathbf{Z},\mathbf{D}} q_w(\mathbf{Z}, \mathbf{D} \mid \mathbf{L}, \mathbf{X}, \mathbf{A}) \log \mathbf{p}(\mathbf{L}, \mathbf{X}, \mathbf{A}) \mathbf{dZdD}
$$
$$
= \int_{\mathbf{Z},\mathbf{D}} q_w(\mathbf{Z}, \mathbf{D} \mid \mathbf{L}, \mathbf{X}, \mathbf{A}) \left( \log \mathbf{p}(\mathbf{D}, \mathbf{Z}, \mathbf{L}, \mathbf{X}, \mathbf{A}) - \log \mathbf{p}(\mathbf{D}, \mathbf{Z} \mid \mathbf{L}, \mathbf{X}, \mathbf{A}) \right) \mathbf{dZdD}.
\tag{3}
$$

On the left side, $\log p(\mathbf{L}, \mathbf{X}, \mathbf{A})$ is independent of $\mathbf{D}$ and $\mathbf{Z}$:

$$
\log p(\mathbf{L}, \mathbf{X}, \mathbf{A}) = \int_{\mathbf{Z},\mathbf{D}} \mathbf{q_w}(\mathbf{Z}, \mathbf{D} \mid \mathbf{L}, \mathbf{X}, \mathbf{A})(\log \mathbf{p}(\mathbf{Z}, \mathbf{D}, \mathbf{L}, \mathbf{X}, \mathbf{A})
$$
$$
- \log p(\mathbf{Z}, \mathbf{D} \mid \mathbf{L}, \mathbf{X}, \mathbf{A}))\mathbf{dZdD}
$$
$$
= \int_{\mathbf{Z},\mathbf{D}} q_{\mathbf{w}}(\mathbf{Z}, \mathbf{D} \mid \mathbf{L}, \mathbf{X}, \mathbf{A}) \left( \log \frac{\mathbf{p}(\mathbf{Z}, \mathbf{D}, \mathbf{L}, \mathbf{X}, \mathbf{A})}{\mathbf{q_w}(\mathbf{Z}, \mathbf{D} \mid \mathbf{L}, \mathbf{X}, \mathbf{A})} - \log \frac{\mathbf{p}(\mathbf{Z}, \mathbf{D} \mid \mathbf{L}, \mathbf{X}, \mathbf{A})}{\mathbf{q_w}(\mathbf{Z}, \mathbf{D} \mid \mathbf{L}, \mathbf{X}, \mathbf{A})} \right) \tag{4}
$$
$$
= \int_{\mathbf{Z},\mathbf{D}} q_{\mathbf{w}}(\mathbf{Z}, \mathbf{D} \mid \mathbf{L}, \mathbf{X}, \mathbf{A}) \log \frac{\mathbf{p}(\mathbf{Z}, \mathbf{D}, \mathbf{L}, \mathbf{X}, \mathbf{A})}{\mathbf{q_w}(\mathbf{Z}, \mathbf{D} \mid \mathbf{L}, \mathbf{X}, \mathbf{A})} \mathbf{dZdD}
$$
$$
+ \mathrm{KL}\left[ q_{\mathbf{w}}(\mathbf{Z}, \mathbf{D} \mid \mathbf{L}, \mathbf{X}, \mathbf{A}) \| \mathbf{p}(\mathbf{Z}, \mathbf{D} \mid \mathbf{L}, \mathbf{X}, \mathbf{A}) \right].
$$

On the right side, the first term is called ELBO:

$$
\mathcal{L}_{ELBO} = \int_{\mathbf{Z},\mathbf{D}} q_{\mathbf{w}}(\mathbf{Z}, \mathbf{D} \mid \mathbf{L}, \mathbf{X}, \mathbf{A}) \log \frac{\mathbf{p}(\mathbf{Z}, \mathbf{D}, \mathbf{L}, \mathbf{X}, \mathbf{A})}{\mathbf{q_w}(\mathbf{Z}, \mathbf{D} \mid \mathbf{L}, \mathbf{X}, \mathbf{A})} \mathbf{dZdD}.
\tag{5}
$$

Then we have:

$$
\log p(\mathbf{L}, \mathbf{X}, \mathbf{A}) = \mathcal{L}_{\mathbf{ELBO}} + \mathrm{KL}\left[ \mathbf{q_w}(\mathbf{Z}, \mathbf{D} \mid \mathbf{L}, \mathbf{X}, \mathbf{A}) \| \mathbf{p}(\mathbf{Z}, \mathbf{D} \mid \mathbf{L}, \mathbf{X}, \mathbf{A}) \right].
\tag{6}
$$

$\mathcal{L}_{ELBO}$ can be calculated as:

$$
\mathcal{L}_{ELBO} = \int_{\mathbf{Z},\mathbf{D}} q_{\mathbf{w}}(\mathbf{Z}, \mathbf{D} \mid \mathbf{L}, \mathbf{X}, \mathbf{A}) \log \frac{\mathbf{p}(\mathbf{Z}, \mathbf{D}, \mathbf{L}, \mathbf{X}, \mathbf{A})}{\mathbf{q_w}(\mathbf{Z}, \mathbf{D} \mid \mathbf{L}, \mathbf{X}, \mathbf{A})} \mathbf{dZdD}
$$
$$
= \mathbb{E}_{q_{\mathbf{w}}(\mathbf{Z},\mathbf{D}|\mathbf{L},\mathbf{X},\mathbf{A})} \left[ \log \frac{p(\mathbf{Z}, \mathbf{D}, \mathbf{L}, \mathbf{X}, \mathbf{A})}{q_{\mathbf{w}}(\mathbf{Z}, \mathbf{D} \mid \mathbf{L}, \mathbf{X}, \mathbf{A})} \right]
$$
$$
= \mathbb{E}_{q_{\mathbf{w}}(\mathbf{Z},\mathbf{D}|\mathbf{L},\mathbf{X},\mathbf{A})} \left[ \log \frac{p(\mathbf{Z})\mathbf{p}(\mathbf{D})\mathbf{p}(\mathbf{L}, \mathbf{X}, \mathbf{A} \mid \mathbf{Z}, \mathbf{D})}{q_{\mathbf{w}_1}(\mathbf{D} \mid \mathbf{L}, \mathbf{X}, \mathbf{A})\mathbf{q_{w_2}}(\mathbf{Z} \mid \mathbf{D}, \mathbf{X})} \right] \tag{7}
$$
$$
= \mathbb{E}_{q_{\mathbf{w}}(\mathbf{Z},\mathbf{D}|\mathbf{L},\mathbf{X},\mathbf{A})[\log \mathbf{p}(\mathbf{L},\mathbf{X},\mathbf{A}|\mathbf{Z},\mathbf{D})]}
$$
$$
+ \mathbb{E}_{q_{\mathbf{w}}(\mathbf{Z},\mathbf{D}|\mathbf{L},\mathbf{X},\mathbf{A})} \left[ \log \frac{p(\mathbf{Z})\mathbf{p}(\mathbf{D})}{q_{\mathbf{w}_1}(\mathbf{D} \mid \mathbf{L}, \mathbf{X}, \mathbf{A})\mathbf{q_{w_2}}(\mathbf{Z} \mid \mathbf{D}, \mathbf{X})} \right].
$$

The first term of $\mathcal{L}_{ELBO}$ can be calculated as:

$$
\mathbb{E}_{q_{\mathbf{w}}(\mathbf{Z},\mathbf{D}|\mathbf{L},\mathbf{X},\mathbf{A})[\log \mathbf{p}(\mathbf{L},\mathbf{X},\mathbf{A}|\mathbf{Z},\mathbf{D})]} = \mathbb{E}_{q_{\mathbf{w}}(\mathbf{Z},\mathbf{D}|\mathbf{L},\mathbf{X},\mathbf{A})}[\log p(\mathbf{X} \mid \mathbf{Z}, \mathbf{D})]
$$
$$
+ \mathbb{E}_{q_{\mathbf{w}}(\mathbf{Z},\mathbf{D}|\mathbf{L},\mathbf{X},\mathbf{A})}[\log P(\mathbf{L} \mid \mathbf{D})] \tag{8}
$$
$$
+ \mathbb{E}_{q_{\mathbf{w}}(\mathbf{Z},\mathbf{D}|\mathbf{L},\mathbf{X},\mathbf{A})}[\log p(\mathbf{A} \mid \mathbf{D})].
$$

The second term of $\mathcal{L}_{ELBO}$ can be calculated as:

$$
\mathbb{E}_{q_{\mathbf{w}}}(\mathbf{Z}, \mathbf{D} \mid \mathbf{L}, \mathbf{X}, \mathbf{A})[\log \mathbf{p}(\mathbf{A} \mid \mathbf{D})]
$$
$$
= \mathbb{E}_{q_{\mathbf{w}_1}(\mathbf{D}|\mathbf{L},\mathbf{X},\mathbf{A})} \mathbb{E}_{q_{\mathbf{w}_2}}(\mathbf{Z} \mid \mathbf{D}, \mathbf{X}) \left[ \log \frac{\mathbf{p}(\mathbf{Z})\mathbf{p}(\mathbf{D})}{\mathbf{q_{w_1}}(\mathbf{D} \mid \mathbf{L}, \mathbf{X}, \mathbf{A})\mathbf{q_{w_2}}(\mathbf{Z} \mid \mathbf{D}, \mathbf{X})} \right]
$$
$$
= \mathbb{E}_{q_{\mathbf{w}_1}(\mathbf{D}|\mathbf{L},\mathbf{X},\mathbf{A})} \mathbb{E}_{q_{\mathbf{w}_2}(\mathbf{Z}|\mathbf{D},\mathbf{X})} \left[ \log \frac{p(\mathbf{D})}{q_{\mathbf{w}_1}(\mathbf{D} \mid \mathbf{L}, \mathbf{X}, \mathbf{A})} \right] \tag{9}
$$
$$
+ \mathbb{E}_{q_{\mathbf{w}_1}(\mathbf{D}|\mathbf{L},\mathbf{X},\mathbf{A})} \mathbb{E}_{q_{\mathbf{w}_2}(\mathbf{Z}|\mathbf{D},\mathbf{X})} \left[ \log \frac{p(\mathbf{Z})}{q_{\mathbf{w}_2}(\mathbf{Z} \mid \mathbf{D}, \mathbf{X})} \right]
$$
$$
= -\mathrm{KL}\left[ q_{\mathbf{w}_1}(\mathbf{D} \mid \mathbf{L}, \mathbf{X}, \mathbf{A}) \| \mathbf{p}(\mathbf{D}) \right] - \mathrm{KL}\left[ q_{\mathbf{w}_2}(\mathbf{Z} \mid \mathbf{D}, \mathbf{X}) \| \mathbf{p}(\mathbf{Z}) \right].
$$

Table 1: Characteristics of the experimental datasets.

| Dataset | $|\mathcal{S}|$ | $\dim(\mathcal{S})$ | $L(\mathcal{S})$ | Domain |
|---|---|---|---|---|
| CAL500 | 502 | 68 | 174 | Music |
| image | 2000 | 294 | 5 | Images |
| scene | 2407 | 294 | 6 | Images |
| yeast | 2417 | 103 | 14 | Biology |
| corel5k | 5000 | 499 | 374 | Images |
| rcv1-s1 | 6000 | 944 | 101 | Text |
| corel16k-s1 | 13766 | 500 | 153 | Images |
| delicious | 16105 | 500 | 983 | Text |
| iaprtc12 | 19627 | 1000 | 291 | Images |
| espgame | 20770 | 1000 | 268 | Images |
| mirflickr | 25000 | 1000 | 38 | Images |
| tmc2007 | 28596 | 981 | 22 | Text |

Table 2: Predictive performance of each comparing approach (mean±std) in terms of *Hamming loss* ↓. The best performance (the smaller the better) is shown in bold face.

| Datasets | SMILE | AN | AN-LS | WAN | ROLE | GLOCAL | MLML | D2ML |
|---|---|---|---|---|---|---|---|---|
| CAL500 | **0.148±0.000** | 0.148±0.000 | 0.149±0.001 | 0.296±0.007 | 0.148±0.000 | 0.148±0.000 | 0.148±0.000 | 0.148±0.000 |
| image | **0.205±0.008** | 0.216±0.012 | 0.213±0.014 | 0.321±0.050 | 0.214±0.019 | 0.211±0.004 | 0.227±0.005 | 0.712±0.018 |
| scene | **0.124±0.035** | 0.141±0.021 | 0.137±0.023 | 0.193±0.029 | 0.174±0.014 | 0.149±0.017 | 0.174±0.019 | 0.288±0.007 |
| yeast | **0.205±0.003** | 0.306±0.000 | 0.306±0.000 | 0.215±0.003 | 0.213±0.006 | 0.277±0.073 | 0.306±0.035 | 0.694±0.015 |
| corel5k | **0.010±0.000** | 0.010±0.000 | 0.010±0.000 | 0.038±0.002 | 0.010±0.000 | 0.010±0.000 | 0.010±0.000 | 0.020±0.000 |
| rcv1-s1 | **0.027±0.000** | 0.028±0.000 | 0.028±0.000 | 0.047±0.004 | 0.028±0.000 | 0.029±0.000 | 0.029±0.000 | 0.917±0.000 |
| corel16k-s1 | **0.019±0.004** | 0.019±0.000 | 0.019±0.000 | 0.136±0.005 | 0.019±0.000 | 0.019±0.000 | 0.019±0.000 | 0.077±0.000 |
| delicious | **0.019±0.001** | 0.019±0.000 | 0.019±0.000 | 0.075±0.007 | 0.019±0.000 | 0.019±0.000 | 0.019±0.000 | 0.326±0.000 |
| iaprtc12 | **0.019±0.011** | 0.019±0.000 | 0.019±0.000 | 0.195±0.007 | 0.019±0.000 | 0.019±0.000 | 0.019±0.000 | 0.019±0.000 |
| espgame | **0.017±0.003** | 0.017±0.000 | 0.017±0.000 | 0.174±0.009 | 0.017±0.000 | 0.017±0.000 | 0.017±0.000 | 0.017±0.000 |
| mirflickr | **0.118±0.001** | 0.127±0.000 | 0.127±0.000 | 0.211±0.003 | 0.130±0.005 | 0.128±0.000 | 0.128±0.000 | 0.128±0.000 |
| tmc2007 | **0.063±0.000** | 0.085±0.001 | 0.089±0.001 | 0.092±0.004 | 0.065±0.002 | 0.098±0.001 | 0.098±0.001 | 0.098±0.000 |

Then we have:

$$\mathcal{L}_{ELBO} = \mathbb{E}_{q_{\mathbf{w}}(\mathbf{Z},\mathbf{D}|\mathbf{L},\mathbf{X},\mathbf{A})}[\log p(\mathbf{X} \mid \mathbf{Z}, \mathbf{D}) + \log \mathbf{P}(\mathbf{L} \mid \mathbf{D}) + \log \mathbf{p}(\mathbf{A} \mid \mathbf{D})] \\ - \text{KL}\left[q_{\mathbf{w}_1}(\mathbf{D} \mid \mathbf{L}, \mathbf{X}, \mathbf{A})\|\mathbf{p}(\mathbf{D})\right] - \text{KL}\left[q_{\mathbf{w}_2}(\mathbf{Z} \mid \mathbf{D}, \mathbf{X})\|\mathbf{p}(\mathbf{Z})\right]. \tag{10}$$

### A.3 Proof of Lemma 1

In order to prove this lemma, we first show that the one direction $\sup_{f\in\mathcal{F}} R_{sp}(f) - \widehat{R}_{sp}(f)$ is bounded with probability at least $1 - \delta/2$, and the other direction can be similarly shown. Suppose an example $(\boldsymbol{x}, y)$ is replaced by another arbitrary example $(\boldsymbol{x}', y')$, then the change of $\sup_{f\in\mathcal{F}} R_{sp}(f) - \widehat{R}_{sp}(f)$ is no greater than $M/(2n)$, the loss function $\mathcal{L}_{sp}$ are bounded by $M$. By applying McDiarmid's inequality, for any $\delta > 0$, with probability at least $1 - \delta/2$,

$$\sup_{f\in\mathcal{F}} R_{sp}(f) - \widehat{R}_{sp}(f) \leq \mathbb{E}\left[\sup_{f\in\mathcal{F}} R_{sp}(f) - \widehat{R}_{sp}(f)\right] + \frac{M}{2}\sqrt{\frac{\log\frac{2}{\delta}}{2n}}. \tag{11}$$

By sysmmetrization, we can obtain

$$\mathbb{E}\left[\sup_{f\in\mathcal{F}} R_{sp}(f) - \widehat{R}_{sp}(f)\right] \leq 2\widetilde{\mathfrak{R}}_n(\mathcal{G}_{sp}). \tag{12}$$

By further taking into account the other side $\sup_{f\in\mathcal{F}} R_{sp}(f) - \widehat{R}_{sp}(f)$, we have for any $\delta > 0$, with probability at least $1 - \delta$,

$$\sup_{f\in\mathcal{F}} \left|R_{sp}(f) - \widehat{R}_{sp}(f)\right| \leq 2\widetilde{\mathfrak{R}}_n(\mathcal{G}_{sp}) + \frac{M}{2}\sqrt{\frac{\log\frac{2}{\delta}}{2n}}. \tag{13}$$

Table 3: Predictive performance of each comparing approach (mean±std) in terms of *Ranking loss ↓*. The best performance (the smaller the better) is shown in bold face.

| Datasets | SMILE | AN | AN-LS | WAN | ROLE | GLOCAL | MLML | D2ML |
|---|---|---|---|---|---|---|---|---|
| CAL500 | **0.239±0.010** | 0.266±0.045 | 0.391±0.048 | 0.244±0.005 | 0.384±0.010 | 0.366±0.009 | 0.478±0.001 | 0.506±0.013 |
| image | 0.170±0.055 | 0.330±0.092 | 0.325±0.084 | 0.240±0.045 | 0.234±0.034 | 0.179±0.004 | **0.163±0.003** | 0.459±0.014 |
| scene | 0.086±0.045 | 0.170±0.132 | 0.171±0.119 | 0.108±0.014 | 0.163±0.045 | 0.108±0.006 | **0.056±0.007** | 0.383±0.035 |
| yeast | **0.161±0.003** | 0.165±0.002 | 0.168±0.002 | 0.163±0.001 | 0.168±0.001 | 0.332±0.007 | 0.361±0.000 | 0.488±0.007 |
| corel5k | 0.134±0.003 | 0.113±0.001 | 0.189±0.011 | **0.111±0.001** | 0.266±0.013 | 0.139±0.002 | 0.355±0.003 | 0.484±0.001 |
| rcv1-s1 | **0.042±0.000** | 0.046±0.001 | 0.060±0.001 | 0.042±0.000 | 0.071±0.004 | 0.168±0.003 | 0.179±0.007 | 0.437±0.002 |
| corel16k-s1 | **0.133±0.001** | 0.138±0.002 | 0.181±0.002 | 0.134±0.001 | 0.241±0.006 | 0.690±0.001 | 0.306±0.005 | 0.454±0.002 |
| delicious | 0.126±0.000 | 0.133±0.002 | 0.276±0.015 | **0.125±0.001** | 0.306±0.007 | 0.445±0.011 | 0.325±0.004 | 0.456±0.004 |
| iaprtc12 | **0.115±0.002** | 0.128±0.003 | 0.230±0.011 | 0.140±0.005 | 0.167±0.002 | 0.442±0.003 | 0.266±0.011 | 0.502±0.015 |
| espgame | **0.158±0.001** | 0.163±0.006 | 0.268±0.004 | 0.158±0.001 | 0.241±0.006 | 0.464±0.001 | 0.319±0.023 | 0.500±0.003 |
| mirflickr | **0.117±0.002** | 0.118±0.003 | 0.148±0.003 | 0.123±0.002 | 0.155±0.006 | 0.189±0.019 | 0.944±0.003 | 0.496±0.007 |
| tmc2007 | 0.049±0.001 | 0.047±0.001 | 0.060±0.002 | **0.045±0.001** | 0.061±0.002 | 0.144±0.003 | 0.143±0.001 | 0.453±0.001 |

Table 4: Predictive performance of each comparing approach (mean±std) in terms of *Coverage ↓*. The best performance (the smaller the better) is shown in bold face.

| Datasets | SMILE | AN | AN-LS | WAN | ROLE | GLOCAL | MLML | D2ML |
|---|---|---|---|---|---|---|---|---|
| CAL500 | 0.865±0.008 | 0.881±0.014 | 0.937±0.017 | 0.878±0.015 | 0.953±0.012 | 0.875±0.013 | **0.668±0.001** | 0.694±0.003 |
| image | **0.171±0.045** | 0.298±0.075 | 0.294±0.069 | 0.225±0.037 | 0.221±0.028 | 0.177±0.018 | 0.783±0.005 | 0.966±0.014 |
| scene | **0.084±0.037** | 0.155±0.112 | 0.156±0.101 | 0.102±0.012 | 0.146±0.036 | 0.103±0.002 | 0.414±0.002 | 0.931±0.004 |
| yeast | **0.455±0.007** | 0.456±0.008 | 0.469±0.010 | 0.460±0.004 | 0.476±0.004 | 0.689±0.001 | 0.942±0.003 | 0.951±0.002 |
| corel5k | 0.312±0.007 | **0.273±0.002** | 0.447±0.022 | 0.273±0.001 | 0.557±0.025 | 0.328±0.005 | 0.396±0.008 | 0.465±0.016 |
| rcv1-s1 | **0.107±0.001** | 0.117±0.003 | 0.153±0.004 | 0.107±0.000 | 0.177±0.007 | 0.315±0.004 | 0.439±0.002 | 0.731±0.003 |
| corel16k-s1 | **0.269±0.003** | 0.280±0.006 | 0.364±0.005 | 0.271±0.001 | 0.465±0.010 | 0.847±0.007 | 0.740±0.004 | 0.848±0.006 |
| delicious | 0.630±0.002 | 0.647±0.012 | 0.894±0.013 | **0.626±0.003** | 0.910±0.004 | 0.861±0.009 | 0.749±0.019 | 0.829±0.002 |
| iaprtc12 | **0.336±0.003** | 0.361±0.007 | 0.593±0.019 | 0.377±0.010 | 0.446±0.005 | 0.695±0.011 | 0.793±0.007 | 0.934±0.008 |
| espgame | **0.382±0.004** | 0.395±0.017 | 0.603±0.009 | 0.384±0.002 | 0.556±0.012 | 0.721±0.018 | 0.850±0.004 | 0.935±0.006 |
| mirflickr | **0.327±0.003** | 0.328±0.003 | 0.397±0.005 | 0.332±0.002 | 0.396±0.013 | 0.436±0.016 | 0.944±0.012 | 0.990±0.003 |
| tmc2007 | 0.130±0.002 | 0.124±0.002 | 0.149±0.004 | **0.120±0.001** | 0.150±0.004 | 0.264±0.004 | 0.834±0.003 | 0.985±0.000 |

## A.4 Proof of Lemma 2

As $w^j$ and $\bar{w}^j$ are bounded in $[0, \kappa]$, we can obtain $\widetilde{\mathfrak{R}}_n(\mathcal{G}_{spl}) \leq \kappa c \left( \mathfrak{R}_n(\ell \circ \mathcal{F}) + \mathfrak{R}_n(\bar{\ell} \circ \mathcal{F}) \right)$ where $\ell \circ \mathcal{F}$ denotes $\{\ell \circ \mathcal{F} | f \in \mathcal{F}\}$ and $\bar{\ell} \circ \mathcal{F}$ denotes $\{\bar{\ell} \circ \mathcal{F} | f \in \mathcal{F}\}$. Since $\mathcal{H}_y = \{h : \boldsymbol{x} \mapsto f_y(\boldsymbol{x}) | f \in \mathcal{F}\}$ and the loss functions $\ell(f(\boldsymbol{x}), y)$ and $\bar{\ell}(f(\boldsymbol{x}), y)$ are $\rho^+$-Lipschitz and $\rho^-$-Lipschitz with respect to $f(\boldsymbol{x})$ $(0 < \rho^+ < \infty$ and $0 < \rho^- < \infty)$ for all $y \in \mathcal{Y}$, by the Rademacher vector contraction inequality, we have $\mathfrak{R}_n(\ell \circ \mathcal{F}) + \mathfrak{R}_n(\bar{\ell} \circ \mathcal{F}) \leq \sqrt{2}(\rho^+ + \rho^-) \sum_{j=1}^c \mathfrak{R}_n(\mathcal{H}_y)$.

## A.5 Proof of Theorem 1

Combining Lemma 1 and 2, we have

$$
\begin{aligned}
R(\widehat{f}_{sp}) - R(f^*) &= R(\widehat{f}_{sp}) - \widehat{R}_{sp}(\widehat{f}) + \widehat{R}_{sp}(\widehat{f}) - \widehat{R}_{sp}(f^*) + \widehat{R}_{sp}(f^*) - R(f^*) \\
&\leq R(\widehat{f}_{sp}) - \widehat{R}_{sp}(\widehat{f}) + \widehat{R}_{sp}(f^*) - R(f^*) \\
&\leq 2 \sup_{f \in \mathcal{F}} \left| R_{sp}(f) - \widehat{R}_{sp}(f) \right| \\
&\leq 4 \widetilde{\mathfrak{R}}_n(\mathcal{G}_{sp}) + M \sqrt{\frac{\log \frac{2}{\delta}}{2n}} \\
&\leq 4\sqrt{2} \kappa c (\rho^+ + \rho^-) \sum_{j=1}^c \mathfrak{R}_n(\mathcal{H}_y) + M \sqrt{\frac{\log \frac{2}{\delta}}{2n}}.
\end{aligned}
\tag{14}
$$

which concludes the proof.

## A.6 Details of Experiments

Some basic statistics about these datasets are given in Table 1, including the number of examples $(|S|)$, the number of features $(\dim(S))$, and the number of class labels $(L(S))$. Tables 2 to 4 show the results of all approaches on *One-error*, *Hamming loss*, and *Coverage*, respectively. Tables 5

Table 5: Predictive performance of SMILE and its variant (mean±std) in terms of *Hamming Loss* and *Coverage*.

| Datasets | Hamming loss ↓ | | Coverage ↓ | |
|---|---|---|---|---|
| | SMILE | SMILE-SI | SMILE | SMILE-SI |
| CAL500 | **0.148±0.000** | 0.148±0.000 | **0.865±0.008** | 0.897±0.002 |
| image | **0.205±0.008** | 0.229±0.000 | **0.171±0.045** | 0.376±0.007 |
| scene | **0.124±0.035** | 0.169±0.008 | **0.084±0.037** | 0.152±0.030 |
| yeast | **0.205±0.003** | 0.306±0.000 | **0.455±0.007** | 0.457±0.003 |
| corel5k | **0.010±0.000** | 0.010±0.000 | 0.312±0.007 | **0.282±0.001** |
| rcv1-s1 | **0.027±0.000** | 0.029±0.000 | **0.107±0.001** | 0.138±0.001 |
| corel16k-s1 | **0.019±0.004** | 0.019±0.000 | **0.269±0.003** | 0.283±0.000 |
| delicious | **0.019±0.001** | 0.019±0.000 | **0.630±0.002** | 0.663±0.005 |
| iaprtc12 | **0.019±0.011** | 0.019±0.000 | **0.336±0.003** | 0.403±0.000 |
| espgame | **0.017±0.003** | 0.017±0.000 | **0.382±0.004** | 0.412±0.005 |
| mirflickr | **0.118±0.001** | 0.128±0.000 | **0.327±0.003** | 0.335±0.002 |
| tmc2007 | **0.063±0.000** | 0.098±0.000 | 0.130±0.002 | **0.127±0.000** |

shows the results of SMILE and its variant SMILE-SI (mean±std) in terms of *Hamming Loss* and *Coverage*.