# OpenReview forum: "One Positive Label is Sufficient: Single-Positive Multi-Label Learning with Label Enhancement"
_NeurIPS.cc/2022/Conference — NeurIPS 2022 Accept_

### Official Review · Reviewer_dP1Q · 2022-07-05

**Rating:** 8
**Confidence:** 4
**Soundness:** 4 excellent
**Presentation:** 3 good
**Contribution:** 3 good

**Summary:**

This paper considers a cost-effective way on multi-label learning, i.e., single-positive multi-label learning (SPMLL). Theoretically, an unbiased risk estimator is derived and an estimation error bound based on it is established. Practically, a method called SMILE is proposed, which leverages the latent soft labels recovered in a label enhancement process. Experiments on the benchmark datasets validate the effectiveness of the proposed method.

**Questions:**

1. In the real world, if data annotators are required to annotate only one relevant label for multi-label instances, a common phenomenon is that people tend to choose the most common and easy to see label for annotation, while the probability of those uncommon and hidden labels being annotated may be greatly reduced. This fact is somewhat different from the experimental setup in which a positive label of randomly selected instances is used to construct a dataset. How to deal with this difference in practical application? The authors should consider and add reasonable discussion.
2. Noting that the performance of proposed method SMILE on one-error metric is not ideal on image and scene datasets, the authors should add some explanation. As far as I am known, the label cardinality of these two datasets is close to 1. Is this related to this phenomenon?


**Limitations:**

Yes

**Strengths And Weaknesses:**

This paper studies single-positive multi-label learning (SPMLL), which is a cost-effective way to solve the problem of high annotation cost of multi-label learning. The authors propose an unbiased risk estimator of SPMLL, which is shown to be guaranteed to approximately converge to the optimal risk minimizer of fully supervised learning. The authors give in-depth theoretical analysis to derive the error bound to demonstrate its learning consistency.
Extensive experimental results on multiple datasets and evaluation metrics demonstrate the effectiveness of the proposed method.
There are some questions:
1. In the real world, if data annotators are required to annotate only one relevant label for multi-label instances, a common phenomenon is that people tend to choose the most common and easy to see label for annotation, while the probability of those uncommon and hidden labels being annotated may be greatly reduced. This fact is somewhat different from the experimental setup in which a positive label of randomly selected instances is used to construct a dataset. How to deal with this difference in practical application? The authors should consider and add reasonable discussion.
2. Noting that the performance of proposed method SMILE on one-error metric is not ideal on image and scene datasets, the authors should add some explanation. As far as I am known, the label cardinality of these two datasets is close to 1. Is this related to this phenomenon?

---

> ### Author Response · Authors · 2022-08-01
> **Response to Reviewer dP1Q**
>
> Thanks for all your feedback and helpful comments. We agree with you that the in-depth theoretical analysis to derive the error bound to demonstrate the learning consistency is given in this paper. We provide some responses and clarifications.
>
> 1. We agree with you that people might tend to choose the most common and easy to see label for ‘hard’ samples which are hard to completely find all positive labels. On the other hand, people might tend to randomly choose single-label for the other samples in practical applications, which could improve the efficiency of MLL data annotation. Therefore, randomly choosing and most-common choosing are all important in SPMLL in practical application and we will study an instance-dependent generating process of single-positive MLL to deal with the most-common choosing in the future.
>
> 2. The label cardinality of image and scene datasets is close to 1, i.e., most examples in these datasets are not MLL samples. However, the proposed empirical risk estimator adopts binary cross-entropy loss which is appropriate for MLL samples. Therefore, the proposed methods would not be ideal for these two datasets.

---

### Official Review · Reviewer_aB62 · 2022-07-10

**Rating:** 8
**Confidence:** 4
**Soundness:** 3 good
**Presentation:** 3 good
**Contribution:** 3 good

**Summary:**

The paper proposes a novel method for single-positive multi-label learning (SPMLL). The method derives an unbiased risk estimator and establishes a corresponding empirical risk estimator by generating latent soft labels with label enhancement. The authors prove that the proposed method could approximately converge to the optimal risk minimizer of fully supervised multi-label learning, which shows that a single-positive label of each instance is sufficient to train the predictive multi-label model in SPMLL. Experiments on benchmark datasets validate the effectiveness of the proposed method.

**Questions:**

why the proposed method could achieve superior performance against the vanilla variant by approximating d_ij with the confidence of the predictive model?

**Strengths And Weaknesses:**

This paper proposes a novel single-positive multi-label learning method, which derives an unbiased risk estimator and generates the soft labels in the corresponding empirical risk estimator. The authors prove that the proposed method adopted with the unbiased risk estimator is risk-consistent for SPMLL. To the best of my knowledge, this is the first attempt to consider the learning consistency in SPMLL problem. The theoretical results show that training a fine predictive multi-label model on single-positive labeled data is possible, which is vital for the SPMLL setting. I really appreciate the motivation and the theoretical results of the proposed method. This paper generates soft labels via approximating the Beta density parameterized by GCN, which could be optimized by the deduced evidence lower bound. This is novel and effective to estimate the soft label in the empirical risk estimator. The authors analyzed the experiments in detail and the Wilcoxon signed-ranks show that the proposed method achieves significant performance against other methods. Besides, the authors conduct the experiments to validate the helpfulness and the convergence of label enhancement. The paper is clearly written and well organized.

This paper could be further improved. The authors should add more details about learning consistency in section 3 since it is vital for SPMLL. In addition, it would be good to add some more discussion about why the proposed method could achieve superior performance against the vanilla variant by approximating d_ij with the confidence of the predictive model.

---

> ### Author Response · Authors · 2022-08-01
> **Response to Reviewer aB62**
>
> Thanks for all your feedback and helpful comments. We agree with you that our method is novel and effective to estimate the soft label in the empirical risk estimator. We will add more details about the concept of learning consistency in Section 3. The proposed label enhancement process leverages the topological information and approximates the Beta density of soft labels, which could achieve superior performance against the vanilla variant with the confidence of the predictive model.

---

### Official Review · Reviewer_rmev · 2022-07-11

**Rating:** 8
**Confidence:** 5
**Soundness:** 4 excellent
**Presentation:** 4 excellent
**Contribution:** 4 excellent

**Summary:**

Nowadays, the weakly-supervised problem has become more and more popular since the difficulty in annotation acquiring process. Single-positive multi-label learning, where each example is annotated with only one relevant label, can draw the attention of the AI audience. This paper proposes a novel single-positive multi-label learning method via adopting an unbiased risk estimator enhancing and recovering the latent soft label as a label enhancement process. In addition, this paper establishes an estimation error bound that guarantees the learning consistency of the proposed method and proves that the risk minimizer could converge to the Bayes risk minimizer of fully supervised learning. Extensive experiments are conducted and show significant results.

**Questions:**

1.How to calculate the second part of Eq. (9) analytically?

**Limitations:**

The paper has addressed the limitations that the employment of data annotators might be decreased which could lead to a negative societal impact.

**Strengths And Weaknesses:**

This paper for the first time proposes an unbiased risk estimator for single-positive multi-label learning and shows that one can successfully learn a theoretically grounded multi-label classifier for the single-positive multi-label learning problem, which is novel and solid. I think this point motivates the novel solution of solving the single-positive multi-label learning problem by recovering the latent soft label as a label enhancement process. The latent soft label is recovered by an inference model and the topological structure in the data, where the posterior density of the latent soft label is inferred via taking on the approximate Beta density and an evidence lower bound for the optimization is deduced. This is novel and interesting. The established estimation error bound guarantees the learning consistency of the proposed method and demonstrates that the obtained risk minimizer would converge to the optimal risk minimizer of fully supervised learning as the number of training data tends to infinity. According to the experiment results, the proposed method achieves superior performance to other comparing methods. The paper is very well organized and easy to follow. The paper can be improved by refining some problems:
1.	The authors should give more details about how to analytically calculate the second part of Eq. (9).
2.	The paper should add more details about label enhancement in related work.
3.	The paper could show the recovered soft label matrix D over all training examples could converge on other datasets.
4.	The charts and figures in the paper take up too much space (like Table 5, Figure 2 and Figure 3), the authors should try to adjust them.

---

> ### Author Response · Authors · 2022-08-01
> **Response to Reviewer rmev**
>
> We'd like to thank you for your careful readings and helpful comments. We agree with you that this paper is first time proposes an unbiased risk estimator for single-positive multi-label learning and shows that one can successfully learn a theoretically grounded multi-label classifier for the single-positive multi-label learning problem. We provide some responses and clarifications.
> 1.	We will add the calculation of the second part of Eq. (9) in the appendix.
> 2.	 Label enhancement (LE) is a process to recover the label distributions from the training examples. Graph-Laplacian-based LE method constructs a local similarity matrix to preserve the structure of the feature space and transfers logical labels into label distributions with the local similarity matrix. The label propagation technique is employed in to propagate labeling-importance information and generate the label distributions. Manifold base LE approach adopts the locally linear embedding technique to achieve identified label distributions. Tang proposes a low-rank representation LE method via capturing the global	relationships of samples and predicting the implicit label correlation. Zhu adopts the structural information between instances and the privileged information to recover label distributions. A bi-directional loss function is proposed to fully explore the relationship between the feature space and the label distribution space. We will add the details about label enhancement in related work.
> 3.	We will add the recovered soft label matrix D over all training examples on other datasets in the revision.
> 4.	We will adjust the charts and figures in the revision.

---

### Official Review · Reviewer_JbLM · 2022-07-11

**Rating:** 6
**Confidence:** 3
**Soundness:** 3 good
**Presentation:** 3 good
**Contribution:** 3 good

**Summary:**

This paper studies the Single-Positive Multi-Label Learning problem in which the learner only observes one label in the label set of each instance. Based on the estimated unobserved labels for each instance, the authors rewrite the expected risk to obtain an unbiased estimator for the SPMLL problem. The authors propose a label enhancement process to estimate the unobserved labels by enforcing that the recovered label should not be completely different from the confidence estimated by the current prediction. Experiments show the effectiveness of the proposed approach.

**Questions:**

Is there any analysis that how the correctness of the recovered unobserved labels affects the estimation error bound? It is also suggested to provide some experimental results for the accuracy of recovered unobserved labels.

**Limitations:**

The authors have adequately addressed the limitations and potential negative social impact of their work.

**Strengths And Weaknesses:**

Strengths
+ This paper studies an interesting but critical problem, which appears in many real-world applications.
+ The proposed label enhancement process seems to give a nice approximation for the unobserved labels in the SPMLL problem.
+ The empirical studies seem convincing and show the effectiveness of their proposed method.

Weaknesses
- The proposed theoretical analysis is established on the assumption that the proposed label enhancement process accurately recovers the unobserved labels. Based on this premise, the theoretical analysis seems standard and lacks new insights.

---

> ### Author Response · Authors · 2022-08-01
> **Response to Reviewer JbLM (1/2)**
>
> Thanks for all your feedback and helpful comments.
>
> We investigated single-positive multi-label learning (SPMLL) and derived an unbiased risk estimator for SPMLL, which shows that one can successfully learn a theoretically grounded multi-label classifier for the problem. To our best knowledge, it is the first attempt to investigate the learning consistency of SPMLL and demonstrate that the obtained risk minimizer would approximately converge to the optimal risk minimizer of fully supervised learning.
>
> To show the accuracy of the soft labels recovered by the proposed label enhancement process, we have studied the recovering experiments. As there is no groud-truth soft label in MLL datasets, we adopts 13 soft labelled datasets in the recovering  experiments. These datasets [1] contain real soft labels and the corresponding positive labels. Specifically, the single-positive label of each example is randomly chosen from the positive label set. Specifically, the soft labels are recovered from the single-positive labelled data via the proposed label enhancement process. Then, the output soft labels are compared with the ground-truth ones. According to Geng's suggestion [1], three evaluation metrics including Chebyshev distance, Kullback-Leibler divergence, and cosine coefficient are selected to quantify the quality of soft labels. The first two are distance measures and the last one is a similarity measure. Four baseline soft-label-recovering algorithms including ML [2], GLLE [3], LESC [4] and PLEML [5] are employed for comparative studies.
>
> Table 1, Table 2 and Tabel 3 illustrate the performance of the proposed label enhancement method against four baseline algorithms in terms of three evaluation metrics. For each evaluation metric, $\uparrow$ denotes the larger the better and $\downarrow$ denotes the smaller the better. These results show that the proposed method is effective to recover soft labels for single-positive labelled examples.
>
> [1] Label Distribution Learning, IEEE TKDE, 2016
> [2] Multi-label manifold learning, AAAI, 2016
> [3] Label enhancement for label distribution learning, IEEE TKDE, 2021
> [4] Label enhancement with sample correlations via low-rank representation, AAAI, 2020.
> [5] Privileged label enhancement with multi-label learning, IJCAI, 2020
>
> Table 1. Recovery performance on measured by Chebyshev distance $\downarrow$
>
>
>
> |	|ML	|GLLE	|LESC	|PLEML	|Ours|
> | ---- | ----| ---- | ---- | ----| ---- |
> |SJAFFE	|0.8424 	|0.3269 	|0.3069 	|0.3255 	|$0.3024$|
> | Yeast_spoem	| 0.6279 	| 0.1381 	| 0.1347 	| 0.1740 	|  $0.1059 $ |
> | Yeast_spo5	| 0.8681 	| 0.2106 	| 0.1983 	| 0.1837 	| $ 0.1688 $|
> | Yeast_dtt	| 1.1018 	| 0.1753 	| 0.1340 	| 0.1045 	|  $0.0867 $|
> | Yeast_cold	| 1.1249 	| 0.1772 	| 0.1692 	| 0.1541 	|  $0.1290 $|
> | Yeast_heat	| 1.4715 	| 0.2295 	| 0.2219 	| 0.1837 	|$0.1731 $|
> | Yeast_spo	| 1.4824 	| 0.3138 	| 0.2777 	| 0.2559 	|$0.2402 $|
> | Yeast_diau	| 1.8100 	| 0.2981 	| 0.2555 	| 0.2208 	|$0.2066 $|
> | Yeast_elu	| 2.8422 	| 0.2894 	| 0.2673 	| 0.2032 	|$0.1945 $|
> | Yeast_cdc	| 3.0592 	| 0.2961 	| 0.2928 	| 0.2189 	|$0.2084 $|
> | Yeast_alpha	| 3.4782 	| 0.3192 	| 0.3060 	| 0.2140 	|$0.2049 $|
> | SBU_3DFE	| 1.0438 	| 0.4022 	| 0.3978 	| 0.3633 	|$0.2573 $|
> | Movie	| 1.6739 	| 0.6619 	| 0.6919 	| 0.7387 	|$ 0.5735$ |
>
>
> //
>
> Table 2. Recovery performance on measured by Kullback-Leibler divergence $\downarrow$
> |	|ML	|GLLE	|LESC	|PLEML	|Ours|
> | ---- | ----| ---- | ---- | ----| ---- |
> |SJAFFE	|0.2922 	|0.0428 	|0.0342 	|0.0419 	|$0.0313 $|
> |Yeast_spoem	|0.2459 	|0.0310 	|0.0296 	|0.0451 	|$0.0201 $|
> |Yeast_spo5	|0.3957 	|0.0402 	|0.0359 	|0.0293 	|$0.0257 $|
> |Yeast_dtt	|0.5775 	|0.0192 	|0.0115 	|0.0069 	|$0.0051 $|
> |Yeast_cold	|0.5722 	|0.0194 	|0.0178 	|0.0147 	|$0.0109 $|
> |Yeast_heat	|0.7535 	|0.0202 	|0.0192 	|0.0128 	|$0.0115 $|
> |Yeast_spo	|0.7286 	|0.0402 	|0.0312 	|0.0271 	|$0.0239 $|
> |Yeast_diau	|0.9474 	|0.0294 	|0.0217 	|0.0156 	|$0.0139 $|
> |Yeast_elu	|1.3991 	|0.0133 	|0.0114 	|0.0064 	|$0.0058 $|
> |Yeast_cdc	|1.5175 	|0.0131 	|0.0130 	|0.0073 	|$0.0066 $|
> |Yeast_alpha	|1.6931 	|0.0127 	|0.0117 	|0.0057 	|$0.0052 $|
> |SBU_3DFE	|0.3614 	|0.0708 	|0.0691 	|0.0573 	|$0.0271 $|
> |Movie	|0.6629 	|0.1841 	|0.2019 	|0.1884 	|$0.0952 $|

---

> > ### Author Response · Authors · 2022-08-02
> > **Response to Reviewer JbLM (2/2)**
> >
> > Table 3. Recovery performance on measured by cosine coefficient $\uparrow$
> >
> > |	|ML	|GLLE	|LESC	|PLEML	|Ours|
> > | ---- | ----| ---- | ---- | ----| ---- |
> > | SJAFFE	| 0.8039 	| 0.9590 	| 0.9713 	| 0.9609 	|$ 0.9729 $|
> > | Yeast_spoem|0.8721 	|0.9747 	|0.9760 	|0.9627 	|$0.9835 $|
> > | Yeast_spo5	|0.7901 	|0.9670 	|0.9705 	|0.9742 	|$0.9777 $|
> > | Yeast_dtt	|0.6849 	|0.9818 	|0.9892 	|0.9935 	|$0.9952 $|
> > | Yeast_cold	|0.7005 	|0.9817 	|0.9834 	|0.9862 	|$0.9899 $|
> > | Yeast_heat	|0.6230 	|0.9812 	|0.9821 	|0.9878 	|$0.9891 $|
> > | Yeast_spo	|0.6456 	|0.9642 	|0.9723 	|0.9747 	|$0.9778 $|
> > | Yeast_diau	|0.5716 	|0.9734 	|0.9808 	|0.9856 	|$0.9873 $|
> > | Yeast_elu	|0.4380 	|0.9873 	|0.9892 	|0.9938 	|$0.9944 $|
> > | Yeast_cdc	|0.4197 	|0.9876 	|0.9877 	|0.9930 	|$0.9937 $|
> > | Yeast_alpha	|0.3887 	|0.9877 	|0.9887 	|0.9945 	|$0.9950 $|
> > | SBU_3DFE	|0.8049 	|0.9270 	|0.9277 	|0.9445 	|$0.9746 $|
> > | Movie	|0.7676 	|0.9013 	|0.8911 	|0.8771 	|$0.9475 $|

---

### Meta-Review · Area_Chair_5uVY · 2022-08-25

**Recommendation:** Accept
**Confidence:** Certain

**Metareview:**

This paper studies the single-positive multi-label learning problem, in which each example is annotated with only one relevant label. The problem is practical and challenging. To address this problem, this paper proposes a new unbiased estimator with a theoretical guarantee. The idea is novel and technically sound. The experimental results also demonstrate the effectiveness of the proposal. All reviewers agree that this study is novel and solid. So I recommend acceptance.

**Award:**

No

---

### Decision · Program_Chairs · 2022-09-14

Accept